

# Ångström exponent errors prevent accurate visibility measurement

Hengnan Guo, Zefeng Zhang*, Lin Jiang, Junlin An, Bin Zhu, Hanqing Kang, Jing Wang

Key Laboratory for Aerosol-Cloud-Precipitation of China Meteorological Administration, Nanjing University of Information
Science & Technology, Nanjing, 210044, China

*Correspondence to*: Zefeng Zhang (zhangzf01@vip.163.com)

**Abstract.** Visibility is an indicator of atmospheric transparency and is widely used in many research fields, including air
pollution, climate change, ground transportation, and aviation. Although efforts have been made to improve the performance
of visibility meters, a significant error exists in measured visibility data. This study conducts a well-designed simulation
calibration of visibility meters, which proves that current methods of visibility measurement include a false assumption, which
leads to the long-term neglect of an important source of visibility error caused by erroneous values of Ångström exponents.
This error has two characteristics. (1) Independence, in which the magnitude of the error is independent of the performance of
the visibility meter. It is impossible to reduce this error by improving the performance of visibility meters. (2) Uncertainty, in
which the magnitude of the error does not show a clear change pattern but can be substantially larger than the measurement
error of visibility meters. It is impossible to accurately estimate the magnitude of this error nor its influence on visibility
measurements. Our simulations indicate that, as errors in visibility caused by erroneous values of Ångström exponents are
inevitable using current methods of visibility measurement, reliable visibility data cannot be obtained without major
adjustments to current visibility measurement methods.

## 1 Introduction

Visibility is a fundamental meteorological parameter, which is widely used in research into synoptic meteorology, air quality,
climatology, and human health, as well as fields closely related to daily life, such as ground transportation, aviation, and
navigation (Che et al., 2007; Huang et al., 2009; Hyslop, 2009; Konstantopoulos et al., 2010; Li et al., 2016). The performance
of visibility meters has been significantly improved through considerable engineering efforts; however, reliable visibility data
are still not available (Singh et al., 2017). Therefore, it is necessary to discuss whether there are problems with visibility
measurement methods leading to the neglect of important potential sources of errors.

Traditionally, visibility measurements were performed by trained human observers (Watson, 2002). In 1924, Koschmieder
related visibility (v) and atmospheric extinction coefficient (b) at a given contrast threshold (ε) (Eq. (1)), which provided a
theoretical basis for measuring visibility with instruments (Koschmieder, 1924). In 1957, the World Meteorological





Organization (WMO) set a benchmark for visibility measurement by introducing the Meteorological Optical Range (MOR)

(WMO, 1957). Subsequently, automated visibility meters gradually replaced human observations.

$$v = -\frac{\ln \varepsilon}{b} \qquad (1)$$

However, it is vital to determine if the MOR is consistent with human observations of visibility. After deducting the influence

of the measurement error, two main differences exist between the two. First, the contrast threshold ($\varepsilon$) is different. The contrast

threshold of the MOR is defined as a constant value (0.05), whereas that of the human eye is typically between 0.0077 and

0.06 (WMO, 2018), as determined by the physiological structure. Second, the measurement wavelength is different. The MOR

requires a light source at a colour temperature of 2700 K (WMO, 2018), whose emission spectrum peaks at a wavelength of

1.07 μm according to Wien's displacement law. In contrast, human observation is restricted to the visible light range, and is

most sensitive at a wavelength of ~0.55 μm (Feynman et al., 2011). WMO claims that "visibility and MOR should be equal"

if the influence of the contrast threshold can be excluded. Obviously, WMO believes that the choice of measurement

wavelength of the light source will not affect the measurement of visibility.

However, Ångström indicated the spectral dependence of visibility as early as 1929 (Ångström, 1929):

$$v_{\lambda_0} = v_{\lambda_1} \left( \frac{\lambda_1}{\lambda_0} \right)^{-q} \qquad (2)$$

where q is the Ångström exponent, which is used to convert the visibility measured at a wavelength of $\lambda_1$ ($v_{\lambda 1}$) to the visibility

at a reference wavelength of $\lambda_0$ ($v_{\lambda 0}$). Erroneous values of q can clearly lead to errors when converting $v_{\lambda 1}$ to $v_{\lambda 0}$. The

relationship between the relative error of visibility (X) and the absolute error of q ($\Delta q$) is shown in Eq. (3), where $\Delta q$ represents

the deviation of erroneous values of q($q_E$) from the true values of q ($q_T$), $v_{\lambda 0}^T$ is the visibility at the reference wavelength of

$\lambda_0$ calculated from the visibility measured at $\lambda_1$ ($v_{\lambda 1}$) using $q_T$ (Eq. (2)), and $v_{\lambda 0}^E$ is calculated using $q_E$. Equation (3) suggests

that any errors in the values of q will appear as errors in the measurement of visibility, as illustrated in Fig. 1. It should be

noted that the magnitude of this error is related not only to the absolute error of q ($\Delta q$), but also to the choice of measurement

wavelength.

$$X = \frac{v_{\lambda_0}^E - v_{\lambda_0}^T}{v_{\lambda_0}^T} = \left( \frac{\lambda_1}{\lambda_0} \right)^{q_T - q_E} - 1 = \left( \frac{\lambda_1}{\lambda_0} \right)^{-\Delta q} - 1 \qquad (3)$$

The question arises whether reliable values of Ångström exponents (q) can be obtained during visibility measurement. If so,

there is no need to consider the influence of Eq. (3) on errors in the visibility measurement; if not, further investigation should

be conducted into the possible range of $\Delta q$ and its effect on the visibility measurement.





WMO gives an affirmative answer to the above question, i.e., there is no need to consider a deviation from the true values of visibility measured at a wavelength of $\lambda_0$ ($V_{\lambda 0}^{T}$) when converting the visibility measured at $\lambda_1$ ($V_{\lambda 1}$) to $\lambda_0$ ($V_{\lambda 0}$) using Eq. (3); therefore, visibility meters with different measurement wavelengths can obtain consistent visibility measurements. If the opinion of the WMO is correct, the following statements are true: (1) the measurement wavelength of visibility meters can be arbitrarily selected because the visibility measured at any wavelength can be mutually converted; (2) the reference visibility can be artificially defined, such as the MOR, because the reliability of visibility data will not be reduced by converting the visibility measured at various wavelengths into the reference wavelength; and (3) multiple visibility benchmarks such as MOR and meteorological visibility by day can be used simultaneously with no problem, in the same way that units such as grams and kilogrammes can be completely substituted to measure mass.

In fact, existing methods of visibility measurement are formulated under the premise that WMO's judgement is correct, with visibility meters with different wavelengths currently in use. However, WMO has not provided a theoretical basis for obtaining reliable values of q. This study conducts a well-designed simulation calibration of visibility meters, which proves that it is impossible to obtain reliable values of q or to determine the magnitude of Δq. Considering the possible range of q in the atmosphere, it can be inferred that errors in visibility caused by erroneous values of q cannot be ignored; therefore, apparent errors exist in the error estimates of current visibility measurements.

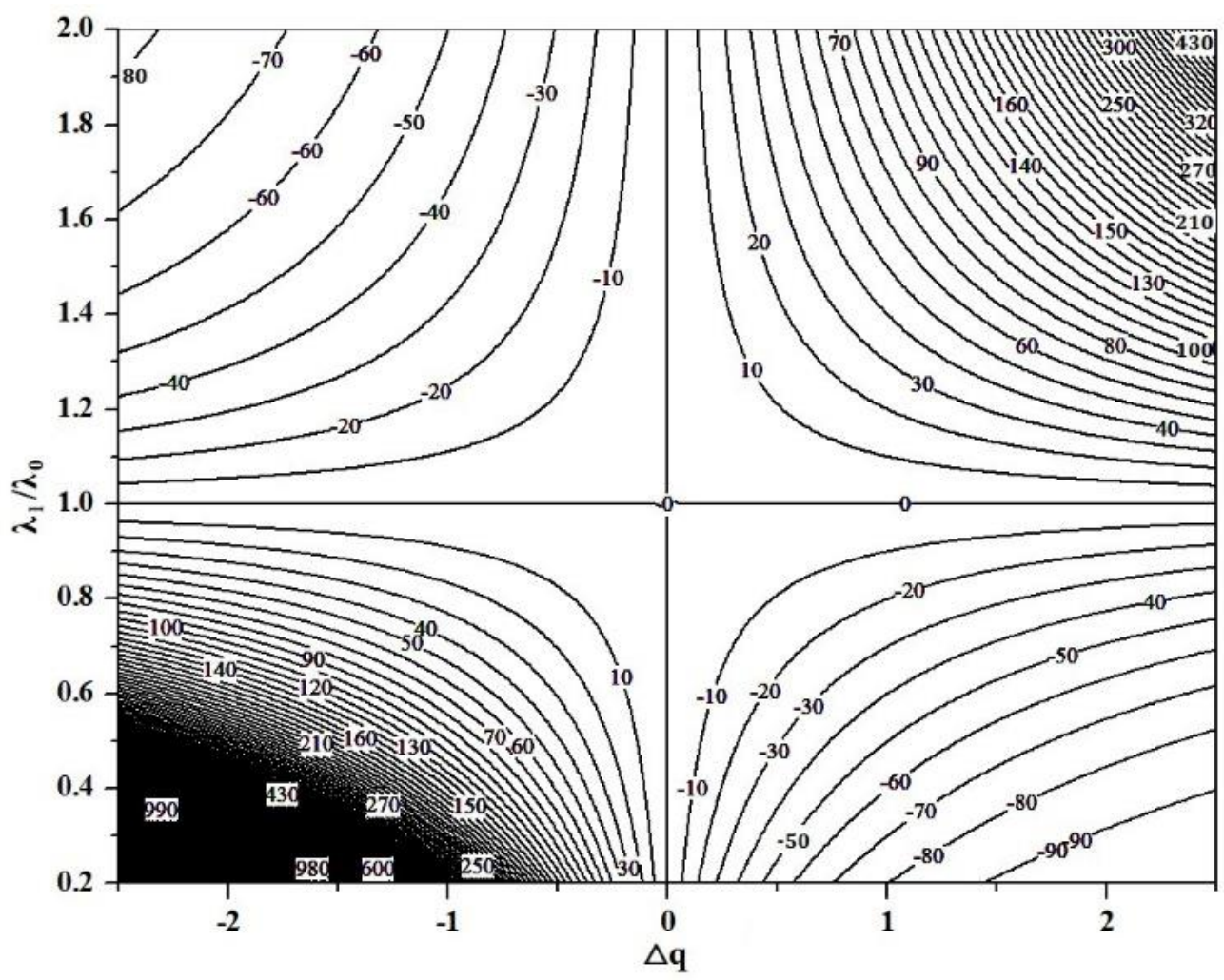

**Figure 1: Influence of the absolute error of the Ångström exponent (q) on the relative error of visibility.**





## 2 Assumptions in the simulation calibration

The aim of the simulation calibration of visibility meters is to develop a function for the Ångström exponent (q), thus

converting the visibility measured at a wavelength of $\lambda_1$ ($v_{\lambda 1}$) to the visibility at the reference wavelength of $\lambda_0$ ($v_{\lambda 0}$). In order

to describe the problem more clearly, the following assumptions are made for the simulation calibration.

(1) The contrast thresholds ($\varepsilon$) of all visibility meters are assumed to be the same. During the calibration, the atmospheric

extinction coefficient (b) was measured without measurement error using ideal visibility meters. Therefore, the values of q can

be accurately calculated in each measurement using Eq. (4), which is obtained by combining Eq. (1) and Eq. (2):

$$-\frac{\ln \varepsilon}{b_{\lambda_0}} = -\frac{\ln \varepsilon}{b_{\lambda_1}}\left(\frac{\lambda_1}{\lambda_0}\right)^{-q} \Rightarrow b_{\lambda_0} = b_{\lambda_1}\left(\frac{\lambda_1}{\lambda_0}\right)^{q} \tag{4}.$$

(2) Assuming that the extinction coefficient (b) can be fully attributed to aerosol particles, and that the information about the

aerosol particles characteristics is clear during the calibration process, the extinction coefficient can be derived using Eq.

(5):

$$b_\lambda = \int \overline{\sigma}_{(D,\lambda,m)} n(D)dD = N\int \overline{\sigma}_{(D,\lambda,m)} f(D)dD \tag{5},$$

where $\overline{\sigma}(D,\lambda)$ accounts for the average extinction coefficient contributed by particles with a diameter of D at an incident-

light wavelength of $\lambda$. The particle size distribution function is given by n(D), where N represents the particle number

concentration and f(D) is the probability distribution function of the aerosol particles, that is, the normalised particle size

distribution function.

$$\left(\frac{\lambda_1}{\lambda_0}\right)^{q} = \frac{b_{\lambda_0}}{b_{\lambda_1}} = \frac{\int_0^{D_{\max}} \overline{\sigma}_{(D,m,\lambda_0)} f(D)\mathrm{d}D}{\int_0^{D_{\max}} \overline{\sigma}_{(D,m,\lambda_1)} f(D)\mathrm{d}D} \tag{6}$$

Equation (6) is derived for calculating the Ångström exponent (q) after combining Eq. (4) and (5). It should be noted that the

particle number concentration (N) is not included in Eq. (6) because it is eliminated in the derivation of Eq. (6), indicating

that q is only related to the physical and chemical parameters of aerosol particles and not to the number concentration.

(3) As for hygroscopic particles on which water vapour condenses, the refractive index of mixed particles can be calculated

using the weighted average of volume ratio of each composition (Jacobson, 2001; Chen et al., 2012; Zhang et al., 2017). The

relationship between the hygroscopic growth factor (GF) and refractive index (m) of mixed particles can be expressed by

Eq. (7), where $m_A$ and $m_w$ represent the refractive indices of dry aerosol particles and water particles, respectively.

$$m = \frac{m_A + m_W(GF-1)}{GF} \tag{7}$$





(4) It is assumed that calibration can be effectively performed by professional calibration personnel and that no subjective errors occur in the calibration process. Calibration personnel know the exact measurement wavelength of the specific visibility meter and can accurately record the measurement results of visibility. During calibration, the measurement results of the reference visibility meter at the reference wavelength of $\lambda_0$ are known to be reliable, unlike those of the visibility meter at $\lambda_1$. Calibration personnel can accurately measure the number concentration of aerosol particles and the ambient relative humidity but cannot measure the physical and chemical parameters of aerosol particles such as the particle size distribution and complex refraction index.

### 3 Simulation of visibility meter calibration

The calibration was conducted simultaneously for four independent groups (urban, marine, rural, and remote continental). The purpose of the calibration was to find a way to convert the visibility measured at a wavelength of $\lambda_1$ (1.07 μm) to the visibility at the reference wavelength of $\lambda_0$ (0.55 μm). During the calibration process, the aerosol particles of the four groups were all homogeneous spherical particles with the same refractive index of 1.53. However, the particle size distribution was not the same but consistent with the typical probability distribution function of aerosol particles in urban, marine, rural, and remote continental, respectively (Seinfeld and Pandis, 2016), as shown in Fig. 2a.

The four groups used exactly the same calibration method and procedure, as follows. First, the number concentration of aerosol particles was set to $N_1$, and the visibility was measured at wavelengths of $\lambda_0$ and $\lambda_1$, recorded as $v_{\lambda 0}^1$ and $v_{\lambda 1}^1$, respectively. The values of q corresponding to $N_1$ were obtained by substituting the measurement results into Eq. (2), which were denoted by $q_1$. Then, the aerosol number concentration was changed and the above steps were repeated. $N_i$ denotes the *i*th change in the aerosol number concentration, the corresponding visibility measurement is $v_{\lambda 0}^i$ and $v_{\lambda 1}^i$, and the calculated q value is $q_i$.

If the aerosol number concentration changed *n* times in the calibration, *n* groups of the values of $v_{\lambda 0}$, $v_{\lambda 1}$, and q were obtained. Finally, these n groups of data were fit to determine the fitting formula for q and eventually determine the calibration function for converting $v_{\lambda 1}$ to $v_{\lambda 0}$.

It can be inferred from the assumptions in the simulation calibration that the calibration data can also be obtained by theoretical calculation, with identical results. Therefore, the calibration process can be calculated and analysed using Mie theory (Bohren and Huffman, 1983). Because only the number concentration of aerosol particles was changed in the simulation calibration, according to Eq. (5) and Eq. (1) and under the assumptions in the calibration, the number concentration of aerosol particles was directly proportional to the extinction coefficient ($b_{\lambda 0}$ and $b_{\lambda 1}$) and inversely proportional to the visibility ($v_{\lambda 0}$ and $v_{\lambda 1}$).

Therefore, for all four groups, q remained constant after substituting n groups of $v_{\lambda 0}$ and $v_{\lambda 1}$ or $b_{\lambda 0}$ and $b_{\lambda 1}$ obtained in the calibration into Eq. (2) or Eq. (6). Then, the calibration function (Eq. (8)) was formulated, where $q_c$ indicates that q is a constant





value. Because of the differences in the size distribution of aerosol particles between groups, the corresponding value of q given by the four groups was bound to be different, as shown in Fig. 2b. Therefore, four different calibration curves were

obtained for the four groups.

$$v_{\lambda_0} = v_{\lambda_1} \left( \frac{\lambda_1}{\lambda_0} \right)^{-q_c}$$  (8)

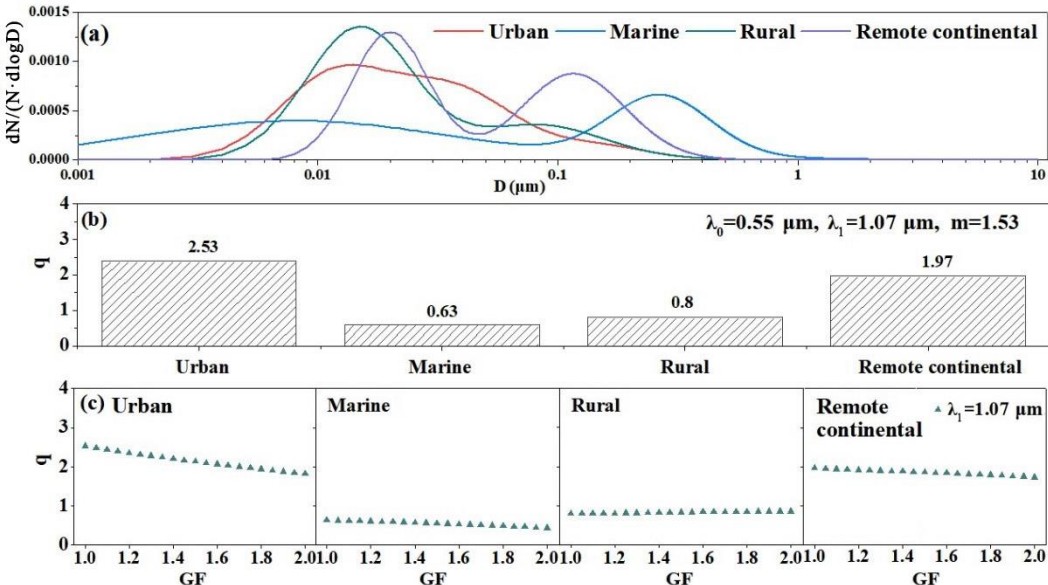

**Figure 2: Critical data used or obtained for the urban, marine, rural and remote continental groups in the simulation calibration:**
**(a) the probability distribution function of aerosol particle size; (b) values of the Ångström exponent (q) obtained in the simulation calibration; and (c) the relationship between q and the hygroscopic growth factor (GF).**

## 4 Discussion on calibration results of visibility meters

### 4.1 Relationship between visibility and the Ångström exponent

The common conclusion reached during calibration of the four groups is that q is a constant variable independent of visibility.
This is a different conclusion from previous studies, where q is determined by an empirical formula that uses visibility as a variable (Middleton, 1957; Kim et al., 2001; Nebuloni, 2005). Therefore, it is vital to determine which conclusion is correct. Equation (6) clearly specifies the determining variables of q, in which the aerosol number concentration does not appear. In the above simulation calibration, visibility is changed by changing the aerosol number concentration. Therefore, we conclude that q is not correlated with visibility. However, a different conclusion would be reached if the visibility is changed by changing
the ambient relative humidity during calibration. Particles absorb water with increasing relative humidity (Mikhailov et al.,



2009; Cheng et al., 2015), causing both the probability distribution function and the refractive index of aerosol particles to change (Eq. 7). Then, it follows from Eq. (5) and Eq. (1) that both the extinction coefficient (b) and visibility (v) are correlated with the relative humidity. The relationship between q and the hygroscopic growth factor (GF) were calculated using Eq. (6) (Fig. 2c). According to Eq. (6), q is a function of the relative humidity; therefore, visibility is correlated with q. We derive an
equation for q as a function of visibility from this calibration for all four groups; however, their functions are different.

In addition to the measurement wavelength, q is determined by the physical and chemical parameters of aerosol particles in Eq. (6). These parameters are constantly changing in the atmosphere; thus, they cannot be directly measured by visibility meters and do not directly correspond to visibility. Regarding equations for q as a function of visibility obtained during calibration or measurement, as long as the physical and chemical parameters of aerosol particles change significantly, the
equations are no longer applicable. Therefore, it is impossible to obtain a universally applicable empirical formula for q and visibility. This raises the inevitable question of why some formulas have been used for such a long time, now that there are no general empirical formulas.

**4.2 Impact of empirical equations on visibility measurement**

The sources of the visibility measurement error can be discussed using Eq. (2). The contrast threshold ($\varepsilon$) is a predefined
constant; therefore, the error of visibility theoretically arises from the measurement error of visibility meters and erroneous values of q. If it is generally believed that a definite function of q and v exists, and that the empirical formula for $q = f(v_{\lambda 1})$ is correct, then Eq. (9) should be written in advance into the programmes of visibility meters whose light source wavelength is $\lambda_1$ rather than the reference wavelength $\lambda_0$, and the output visibility is no longer $v_{\lambda 1}$ but $v'_{\lambda 0}$.

$$v'_{\lambda_0} = v_{\lambda_1} \left( \frac{\lambda_1}{\lambda_0} \right)^{-f(v_{\lambda_1})} \tag{9}$$

For the calibration of visibility meters with a wavelength of $\lambda_1$, the aim is to convert $v'_{\lambda 0}$ to the visibility measured at the reference wavelength ($v_{\lambda 0}$), and the calibration work is assigned to the urban, marine, rural, and remote continental groups. Since they all think that $q = f(v_{\lambda 1})$ is correct, the difference between $v'_{\lambda 0}$ measured by the visibility meter at $\lambda_1$ and $v_{\lambda 0}$ measured at $\lambda_0$ is wrongly attributed to the measurement error. Then, the formula converting $v'_{\lambda 0}$ to $v_{\lambda 0}$ is obtained by this calibration. By comparing Eq. (8) with Eq. (9), Eq. (10) represents the calibration function developed by the four groups,
where $q_c$ denotes the four constants obtained by the four groups, as shown in Fig. 2b.

$$v_{\lambda_0} = v'_{\lambda_0} \left( \frac{\lambda_1}{\lambda_0} \right)^{-q_c + f(v_{\lambda_1})} \tag{10}$$





The calibration function Eq. (10) obtained in this calibration appears to be different from Eq. (8) obtained in the simulation calibration on the surface but is actually the same. The calibration of visibility meters aims to convert the visibility measured at a wavelength of $\lambda_1$ ($v_{\lambda 1}$) to the visibility at the reference wavelength of $\lambda_0$ ($v_{\lambda 0}$). The method of calculating q using a

predefined empirical formula only adds an additional intermediate step, meaning that $v_{\lambda 1}$ is first converted to $v'_{\lambda 0}$ then to $v_{\lambda 0}$ in the calibration; therefore, there is no effect on the final visibility output. As long as the calibration can be effectively completed, the influence of the empirical formula can be theoretically eliminated to achieve identical visibility outputs after calibration, regardless of whether the empirical formula $q = f(v_{\lambda 1})$ is pre-set in the visibility meter, and regardless of the expression of the empirical formula. This conclusion can explain, to a certain extent, why some empirical formulas have

been used to calculate q in visibility measurements for such a long time.

From a purely mathematical perspective, pre-setting an incorrect empirical formula of q has no effect on the visibility outputs, as discussed above. Nevertheless, this does not mean that it has no effect on the error estimation of the visibility measurement, which leads to at least two problems: (1) incorrect attribution of the error and (2) incorrect description of the nature of the error. The first problem is the attribution error, where erroneous values of q are incorrectly attributed to the measurement error

of visibility. Disturbance of the physical and chemical parameters of aerosol particles is inevitable in the actual calibration, leading to changes in the values of q. Therefore, only in Eq. (11), which truly describes the calibration result, does $q_i$ represent the value of q in the $i$th measurement. The first term on the right-hand side of the equation is the empirical formula of q obtained in the simulation calibration and the second term represents the deviation of q in the $i$th calibration ($q_i$) from q calculated using the empirical formula. The measurement errors of instruments can be divided into systematic and random

errors. The attribution error misinterprets the process of obtaining the first term, that is, the empirical formula of q, as the calibration to systematic errors, and misinterprets the second term as random errors.

Second, incorrect attribution of the visibility error will lead to problems describing the nature of the error. Stability is a prerequisite for all measurements. The systematic error is typically a constant deviation from the true value, which can be eliminated by calibration. Random error is not identifiable but follows a certain distribution, such as a normal distribution, and

can be estimated through multiple measurements and calibration. However, q is not directly measured in the current visibility measurement, nor are the physical and chemical parameters determining q. Therefore, the error in selecting the inappropriate q value is not a measurement problem, the first term in Eq. (11) is not a systematic error, and the second term is not a random error. Therefore, it is important to discuss whether existing calibration methods for visibility meters, which are based on the understanding of measurement problems, can reasonably be applied to calibrate the error caused by erroneous values of q. If

the answer is yes, the existing calibration methods for visibility meters are correct; if the answer is no, then a major defect exists in the current measurement methods of visibility meters. The key to answering this question is determining whether the calibration function for visibility meters is generally applicable.

$$q_i = f(v_{\lambda_1}) + \delta q_i \qquad\qquad (11)$$





## 5 Applicability of calibration

The calibration curves of visibility meters were obtained by the urban, marine, rural, and remote continental groups in the simulation calibration. The expressions of the calibration curves all follow Eq. (8); however, the values of $q_c$ are different so the obtained calibration curves are also different. If the error in selecting the inappropriate q value is misunderstood as the measurement error of visibility meters, the cause of this difference is incomprehensible. However, as long as it is understood that q is determined by Eq. (6), it is clear that the difference in the values of $q_c$ is caused by the difference in the probability

distribution function for the size of aerosol particles used by the four groups in the calibration. If visibility is measured by the marine, rural, and remote continental groups after calibration, in parallel with the visibility meter calibrated by the urban group, the relative error of the inter-comparison measurement will be 254%, 213%, and 45%, respectively. Such a large error is caused by erroneous values of q. The WMO provides the following suggestions for this unacceptable error.

According to WMO, "the calibration should be verified regularly in very good visibility, that is, over 10 to 15 km". The WMO

also requires that "atmospheric conditions resulting in erroneous calibration must be avoided" (WMO, 2018). The recommendations of WMO essentially reduce the influence of q on visibility measurement by artificially controlling the calibration conditions. Generally, the physical and chemical characteristics of aerosol particles are stable at specific observation sites. After adding more conditions such as visibility and weather conditions, the variation range of the physical and chemical parameters of aerosol particles will become smaller, and the range of q will become smaller. Obviously,

regardless of whether one visibility meter is calibrated repeatedly or multiple visibility meters are calibrated simultaneously, the calibration result will be approximately the same according to the suggestions of WMO, where the first term in Eq. (11) will be approximately identical and the absolute value of the second term will change minimally. Therefore, as long as the visibility meter results are reliable, the measured visibility data after calibration will exhibit good consistency.

Although the recommendations of WMO effectively solve the consistency problem of visibility measurement data, it cannot

solve the problem of error in visibility caused by erroneous values of q. This is because the physical and chemical characteristics of aerosol particles in the atmosphere are constantly changing. Even at the same observation site, many variables can cause changes in the physical and chemical properties of aerosol particles, such as wind, relative humidity, and emission sources. Clearly, the values of q also change constantly in the atmosphere and will not follow the empirical formula obtained in the calibration. Therefore, we cannot expect a stable term similar to the first term in Eq. (11), and we cannot expect the

second term to have a certain statistical dispersion. As q is not directly or indirectly measured in the current visibility measurement, the actual value of q at the observation site is not clear at the time of observation, nor is the absolute error of q or the error in visibility data caused by erroneous q.





## 6 Estimation of visibility error attributed to q

In order to assess the influence of the Ångström exponent (q) on the error of visibility data, it is necessary to clarify the range
of q in the ambient atmosphere. Although long-term observations of q have not been included in past visibility measurements,
they can be found in abundance in sun photometer measurements. Of course, the measurement objects and principles of sun
photometers are different from those of visibility meters; therefore, the values of q calculated from the aerosol optical depth
(AOD) measured by multi-wavelength photometers might be different from those calculated from visibility data measured by
visibility meters. However, q is determined by the physical and chemical parameters of aerosol particles. In addition, a
number of studies have demonstrated that there is consistency between visibility and AOD measurements (Kaufman and Fraser,
1983; Wu et al., 2014). For example, AOD inversed by visibility agrees well with satellite-based AOD (Wang et al., 2009),
and surface visibility derived from satellite retrievals of AOD exhibits fairly good consistency with measured visibility data
(Kessner et al., 2013). Therefore, it is feasible to discuss the change in q using the values of q derived from AOD data as a
reference.

The Aerosol Robotic Network (AERONET) is the largest global network of photometers with more than 800 stations
worldwide (Holben et al., 1998). We selected 10 sites for analysis, which represent different aerosol types; the details of the
sites are given in Table 1. As the wavelength pair of 0.87/0.44 μm in the measurement wavelengths provided by AERONET
is most similar to the wavelength pair of 1.07/0.55 μm in the visibility measurement, the values of q calculated from AOD at
0.87 μm ($AOD_{0.87}$) and 0.44 μm ($AOD_{0.44}$) are discussed here. The relationship between q and AOD at 0.87 μm is shown in
Fig. 3a, which exhibits no obvious correlation at any measurement site; thus, q cannot be determined by $AOD_{0.87}$. As AOD is
directly proportional to the extinction coefficient of the atmosphere, and the extinction coefficient is inversely proportional to
visibility, it is clear from Fig. 3a that the values of q cannot be determined from visibility even for the same measurement site.
In accordance with the calibration process of visibility meters, we used AOD data at 0.87 μm and 0.44 μm to obtain a scheme
for calculating the Ångström exponent, then determined the function for calculating AOD at 0.44 μm from measured $AOD_{0.87}$
data, and finally evaluated the deviation of calculated values (i.e., AOD data calculated to 0.44 μm) from measured values (i.e.,
measured $AOD_{0.44}$ data). Two evaluation schemes were used. The first performed the calibration once a month, using all q
data of one month to obtain the calibration function for that month. After the calibration was completed, only $AOD_{0.87}$
measurements of that month were used to calculate the AOD at 0.44 μm. Using the measured $AOD_{0.44}$ data as the true value,
we calculated the absolute value of the relative error of calculated AOD data for each specific month over the entire time
period at each site. Figure 3b shows the ratio of the number of data with different absolute values of relative errors according
to scheme 1 to the total number of data for each specific site. It is clear from Fig. 3b that data with large relative errors exist
even if the calibration is performed once a month. The results of scheme 1 indicate that the values of q may change significantly
in a short time. In the second scheme, the calibration was performed monthly using all q data of one month. The difference is
that all measured $AOD_{0.87}$ data over the entire time period were used to calculate the AOD at 0.44 μm. The absolute value of
the relative errors of calculated AOD data for each specific month was calculated over the whole time period then grouped





into bins of the given intervals, as shown in Fig. 3c. Compared to Fig. 3b, the relative error of the calculated AOD at 0.44 μm is far larger using scheme 2 than scheme 1, indicating that a consistent formula for q does not exist, even for the same site. Although the relative error of AOD caused by erroneous values of q cannot be used as a conclusion to evaluate the visibility error caused by erroneous values of q, it can still be used as a reference. The values of q can vary widely in the ambient

atmosphere, and the absolute error of q is not clear. Therefore, the visibility error caused by q cannot be ignored, and sometimes is much larger than that caused by the error in visibility measurement.

**Table 1: Details of AERONET sites used to obtain Ångström exponents (q) and AOD measurements.**

| Site | Aerosol type | Location | Time period | Length of data (in hours) |
|---|---|---|---|---|
| GSFC | Urban industrial | 38°N, 76°W | 5/1993–4/2018 | 157682 |
| Lille | | 50°N, 3°E | 6/1995–10/2018 | 75414 |
| Mongu | Biomass | 15°S, 23°E | 6/1997–2/2010 | 91199 |
| Skukuza | | 24°S, 31°E | 7/1998–5/2018 | 73971 |
| Banizoumbou | Dust | 13°N, 2°E | 10/1995–8/2018 | 170540 |
| Solar Village | | 24°N, 46°E | 2/1999–6/2013 | 168117 |
| Beijing | Mixed | 39°N, 116°E | 3/2001–6/2017 | 88509 |
| Chen-Kung Univ | | 22°N, 120°E | 2/2002–10/2018 | 47471 |
| Mauna Loa | Maritime | 19°N, 155°W | 6/1994–6/2018 | 266209 |
| Midway Island | | 28°N, 177°W | 1/2001–2/2015 | 32960 |






**Figure 3: (a) Relationship between the Ångström exponent (q) and AOD at a wavelength of 0.87 μm at 10 typical sites. (b) and (c) Distribution of absolute value of relative error of AOD data converted to a wavelength of 0.44 caused by erroneous values of q obtained in the calibration using scheme 1 and scheme 2, respectively.**





## 7 Conclusions and recommendations

The MOR is the visibility measurement benchmark set by WMO, but its measurement wavelength is neither the same as the wavelength of maximum sensitivity of the human eye nor the measurement wavelength of common visibility meters. If the absolute error of q can be guaranteed to be small, there will be no problem; otherwise, additional errors will occur in visibility measurements. The calibration simulations performed in this study indicate that it is impossible not only to obtain reliable Ångström exponents but also to determine the visibility error caused by erroneous values of q using current visibility

measurement methods. Considering the wide range of values of q in the ambient atmosphere, the error of visibility caused by erroneous values of q can be much larger than that caused by the measurement error. Therefore, it is impossible to reduce the visibility error by improving the performance of visibility meters alone. Further work is required to improve the measurement accuracy of visibility.

It is also recommended to check historical visibility data. Because the error of visibility caused by erroneous values of q is

ignored, the error of visibility data obtained in the measurement may be much larger than the error given by the manufacturers of visibility meters. Therefore, it is necessary to check previous visibility data and obtain visibility measurements corresponding to specific measurement wavelengths for use in future research.

Further, we recommend that the measurement standards of visibility be modified in order to eliminate the visibility error caused by erroneous values of q. Two ideas are considered here. One is to regulate the measurement wavelength of visibility meters

to be strictly consistent with the reference measurement wavelength. Its advantage is that consistent, reliable, and guaranteed visibility data could be obtained. However, we should be highly cautious regarding the choice of reference measurement wavelength in this case. For example, the MOR requires a light source at a colour temperature of 2700 K, whose measurement wavelength is completely different from that of the human eye. Therefore, as the light emitted by a black body does not consist of a single wavelength, can such a light source be suitable and widely used in visibility measurement? The second idea is to

stop using visibility measured at a specific wavelength as the benchmark for visibility measurement. Instead, we should establish measurement standards focussed on the reliability of the measurement of the extinction coefficient and set industry standards according to different usage scenarios.

### Code and Data availability

The Ångström exponent and AOD data are available at Aerosol Robotic Network (http://aeronet.gsfc.nasa.gov/). The
simulation results and the computer code used here are available on request.

### Author contribution

Zefeng Zhang conceived and designed the experiments and contributed to the analysis of the results. Hengnan Guo performed the simulations and generated the figures. Lin Jiang processed the data and generated the figures. Junlin An, Bin Zhu, Hanqing



Kang, and Jing Wang involved in the discussions and helped in the data analysis. Zefeng Zhang wrote the paper with the contribution of Hengnan Guo.

**Competing interests**

The authors declare that they have no conflict of interest.

**Acknowledgements**

This work was supported jointly by the National Key Research and Development Program of China (2016YFA0602003). We thank the AERONET principal investigators and their staff for establishing and maintaining the sites used in this study.

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
