# Peer review of "Ångström exponent errors prevent accurate visibility measurement"

_Atmospheric Measurement Techniques, 2020_

## Referee Comment (RC1) · Anonymous Referee #1 · 21 Dec 2020

In this manuscript the authors identify an error in determining visibility conditions from "visibility meters." The main focus of the manuscript is that the WMO recommends monitoring visibility at a wavelength of 1.07 um, while visibility occurs at shorter wavelengths (550 nm is the reference assumed). The fundamental issue is that without knowledge of the Angstrom exponent, which is rarely available, it is impossible to accurately correct the measurements into the visible portion of the spectrum. The authors go on to quantify the impacts of this issue on errors in visibility measurements and ultimately suggest that the current approach be changed to deal with these problems. The results are what one might expect, but the authors do a thorough job of quantifying the errors associated with the current methods. I recommend the manuscript be published when the following comments are addressed.

Comments In several places throughout the manuscript, the authors ascribe a belief, opinion or intent to the WMO. This is not appropriate, and the language should be changed. For any large scale endeavor, assumptions and simplifications must be made, and I am guessing that is the case here as well. • Lines 40-41. "Obviously, WMO believes that. . ." should be changed to, "This statement implies that. . ." or something similar. • Lines 56-64. The authors are taking a simplifying assumption from the WMO and suggesting that they don't understand what the Angstrom exponent is. This paragraph should be rewritten to simply state what the implications are. For example, remove phrases like, "If the opinion of the WMO is correct. . ."; • Line 65. ". . .under the premise that WMO's judgement is correct," should be changed to, "WMO's recommended approach is adequate." Or something similar.

It would be worth providing a paragraph describing the primary method(s) that are used as "visibility meters", the wavelengths used, and typical corrections (e.g. at lines 65-66). For research purposes, many current visibility measurements use multiple wavelengths and so the Angstrom exponent can be directly determined. I am not as familiar with the measurements the authors are describing, and it would benefit the reader to have more background information. On this note, where do most measurements fall in relation to the contours of Figure 1?

Section 3 and Figure 2. The authors fix the size distributions and refractive index for four expected aerosol types, and then use these data to determine the Angstrom exponent by varying N (although it is not a function of N). The authors should vary the size distributions and refractive indices within physically reasonable bounds to determine ranges of values for their calculated Angstrom exponents (Figure 2b) for the different aerosol types. This would be a useful result in and of itself.

Section 4.2. The authors note, "If it is generally believed that a definite function of q and v exists. . ." Both q and v depend on aerosol composition, size distributions, hygroscopicity, etc. There is no believing in this relationship – there are reasons the two are related, but simplifying assumptions attempt to relate them in a way that is easy

to implement.

Section 6. In addition to the schemes explored, it would be interesting to determine q from the two wavelengths used (0.87 um and 0.44 um) for every data point, and then determine AOD at a 3rd wavelength from Aeronet (not currently included) collected at the same time, to get a better idea of uncertainty in q due solely to the Aeronet measurements.

Minor Comments Line 45. A statement describing the parameters that affect the Angstrom exponent would be useful here. Line 70. You should be more specific when you state, "...errors exist in the error estimates of current visibility measurements." Research grade instruments do not suffer from these same issues. Line 96. Define what the hygroscopic growth factor is. Lines 139-141. The authors note, "The common conclusion reached during calibration of the four groups is that q is a constant variable independent of visibility. This is a different conclusion from previous studies, where q is determined by an empirical formula that uses visibility as a variable." Other work which assumes a dependence of q (a proxy for size distribution) on visibility (a proxy for aerosol loading) are based on the assumption that greater aerosol loadings typically occur following atmospheric processing (e.g. Pitchford et al., 2007 JAWMA), which results in larger aerosol sizes (and therefore a change in q). This is based on ambient measurements, not simulations where size distributions can be kept constant. Line 167. Who is "thinking" in this sentence. Please re-phrase.

---

## Referee Comment (RC2) · Anonymous Referee #2 · 23 Dec 2020

Guo et al.: **Ångström exponent errors prevent accurate visibility measurement**, Atmos. Meas. Tech. Discuss. https://doi.org/10.5194/amt-2020-415, in review, 2020.

**Review**

**General**

The paper presents theoretical background of visibility measurements and points out systematic errors in essentially all visibility measurements. The theoretical background of the errors is pointed out on lines 34 – 64 of the discussion paper. The explanation is convincing, I really learned new things in reading it. It is obvious that there are systematic errors in visibility measurements worldwide. The topic is definitely important, not only to the scientific community but also to a wider audience: visibility measurements for instance at airports, harbours and at sea are relevant to practically everybody.

The paper is important and basically well written and I can recommend publishing it in AMT.
However, before that I wish you would do some modifications.

First, I can see in your Fig. 2 that there were bimodal size distributions in your simulations already but you did not really pay any attention to it. Bimodality of size distributions have a strong effect on the Ångström exponent, see, e.g.,
Schuster, G. L., Dubovik, O., and Holben, B. N.: Angstrom Exponent and Bimodal Aerosol Size Distributions, J. Geophys. Res., 111, D07207, https://doi.org/10.1029/2005JD006328, 2006.
There are a lot of references in that paper and there are a lot of refs to it that discuss this matter. Among other things it shows that the Ångström exponent often varies with wavelength. And with the ratio of coarse and fine particles. How do these affect your results?

**Detailed comments**

L40 "Obviously, WMO  believes …".  WMO is not a person, it cannot believe anything. Rewrite.

L42 "Ångström indicated the spectral dependence of visibility as early as 1929 (Ångström, 1929) …".
I was wondering, whether Ångström really wrote about visibility and acquired a copy of the original paper. I was right. There is not even the word "visibility" in the whole paper. There is the wavelength dependency of absorption and transmittance and the derivation of the exponent that was later called the Ångström exponent. So, transmittance depends on the Ångström exponent and visibility depends on transmittance according to your Eq. (1). Your Eq. (2) follows from these but it is not given by Ångström (1929).

L59 ".. opinion of the WMO …". again, WMO is not a person, it does not have opinions. Rewrite.

L95-98 "The relationship between the hygroscopic growth factor (GF) and refractive index (m) of mixed particles can be expressed by Eq. (7), where $m_A$ and $m_w$ represent the refractive indices of dry aerosol particles and water particles"

$$m = \frac{m_a + m_w(GF - 1)}{GF} \tag{7}$$

The Eq. (7) is strange, I have never seen it in this form. First, what do you mean with "water particles"? Does it mean humid particles or pure water droplets? Further, GF is generally defined as the ratio of humid and dry particle diameters. So let me develop this further:

$$m = \frac{m_a + m_w(GF - 1)}{GF}; \quad GF = \frac{D_{p,humid}}{D_{p,dry}} \Rightarrow m = \frac{m_a + m_w(GF - 1)}{GF} = \frac{D_{p,dry}}{D_{p,humid}} m_a + \left(\frac{D_{p,humid} - D_{p,dry}}{D_{p,humid}}\right) m_w$$

This is definitely not a volume-weighted average of refractive indices of $m_a$ and $m_w$. So, how was Eq. (7) derived? If it was used like it is written now then you should correct it and repeat your simulations with a corrected one.

---

## Author Comment (AC1) · 17 Feb 2021

Response to reviewers for manuscript amt-2020-415: **Ångström exponent errors prevent accurate visibility measurement**

We thank the editorial team and the reviewers for your valuable inputs in enhancing the quality of our manuscript and appreciate the opportunity provided to revise it for publication in *Atmospheric Measurement Techniques*. We are happy to submit our point-by-point responses to the reviewers' comments and suggestions. The reviewers' comments/suggestions are in black. Our responses are in red. All the suggested changes have been incorporated in the manuscript and are outlined here.

**General Comments:**

In this manuscript the authors identify an error in determining visibility conditions from "visibility meters." The main focus of the manuscript is that the WMO recommends monitoring visibility at a wavelength of 1.07 um, while visibility occurs at shorter wavelengths (550 nm is the reference assumed). The fundamental issue is that without knowledge of the Angstrom exponent, which is rarely available, it is impossible to accurately correct the measurements into the visible portion of the spectrum. The authors go on to quantify the impacts of this issue on errors in visibility measurements and ultimately suggest that the current approach be changed to deal with these problems. The results are what one might expect, but the authors do a thorough job of quantifying the errors associated with the current methods. I recommend the manuscript be published when the following comments are addressed.

**General response:** We thank the reviewer for the valuable review and encouraging comments.

**1. Comment1:** In several places throughout the manuscript, the authors ascribe a belief, opinion or intent to the WMO. This is not appropriate, and the language should be changed. For any large scale endeavor, assumptions and simplifications must be made, and I am guessing that is the case here as well.
Lines 40-41. "Obviously, WMO believes that…" should be changed to, "This statement implies that…" or something similar.
Lines 56-64. The authors are taking a simplifying assumption from the WMO and suggesting that they don't understand what the Angstrom exponent is. This paragraph should be rewritten to simply state what the implications are. For example, remove phrases like, "If the opinion of the WMO is correct…";
Line 65. "…under the premise that WMO's judgement is correct," should be changed to, "WMO's recommended approach is adequate." Or something similar.

**1. Response 1:** We thank the reviewer for this useful comment. We agree that we should have phrased these points differently and not focus on WMO, but rather on specific conclusions presented in the reports by WMO. The suggested changes have been implemented in the manuscript as follows:
In lines 38–40, we have revised the sentence, which now reads as follows: "WMO claims that "visibility and MOR should be equal" if the influence of the contrast threshold can be excluded, which implies that the choice of measurement wavelength of the light source will not affect the measurement of visibility."
In lines 55–73, we have rewritten as follows: The Guide to Instruments and Methods of Observation (WMO, 2018) cites the intercomparison of visibility measurements (Middleton, 1952; WMO, 1990),

where the difference in MOR and visibility by day is attributed to the difference in the contrast threshold. The conclusion reached in this guide states that "Visibility and MOR should be equal if the observer's contrast threshold is 0.05 (using the criterion of recognition) and the extinction coefficient is the same in the vicinity of the instrument, and between the observer and the objects." This conclusion implies that there is no need to consider a deviation from the true values of visibility measured at a wavelength of $\lambda_0$ ($v_{\lambda 0}^{\mathrm{T}}$) when converting the visibility measured at $\lambda_1$ ($v_{\lambda 1}$) to $\lambda_0$ ($v_{\lambda 0}$) using Eq. (3), i.e., the errors in the values of q are negligible. Therefore, visibility meters with different measurement wavelengths can obtain consistent visibility measurements and the following statements are true: (1) the measurement wavelength of visibility meters can be arbitrarily selected because the visibility measured at any wavelength can be mutually converted; (2) the reference visibility can be artificially defined, such as the MOR, because the reliability of visibility data will not be reduced by converting the visibility measured at various wavelengths into the reference wavelength; and (3) multiple visibility benchmarks such as MOR and meteorological visibility by day can be used simultaneously, in the same way that units such as grams and kilogrammes can be completely substituted to measure mass.

In fact, existing methods of visibility measurement using visibility meters with different wavelengths currently in use are formulated under the premise that the WMO's recommended approach is adequate. For example, the light source of Biral RWS-30 is at 850 nm (Biral, 2018), that of Optec LPV-3 is at 550 nm (Optec, 2011), and Vaisala LT31 uses a white light source (Vaisala, 2018). However, this guide does not explain why the MOR is consistent with human observations of visibility when the contrast threshold is the same. In other words, no theoretical basis is provided to prove that reliable values of q can be obtained in the visibility measurement.

The suggested change in line 69 (previously line 65) has been implemented in the manuscript.

**2. Comment 2:** It would be worth providing a paragraph describing the primary method(s) that are used as "visibility meters", the wavelengths used, and typical corrections (e.g. at lines 65-66). For research purposes, many current visibility measurements use multiple wavelengths and so the Angstrom exponent can be directly determined. I am not as familiar with the measurements the authors are describing, and it would benefit the reader to have more background information. On this note, where do most measurements fall in relation to the contours of Figure 1?

**2. Response 2:** As suggested by the reviewer we have added more information on the wavelength of visibility meters as follows (Lines 69–73): "For example, the light source of Biral RWS-30 is at 850 nm (Biral, 2018), that of Optec LPV-3 is at 550 nm (Optec, 2011), and Vaisala LT31 uses a white light source (Vaisala, 2018)."

In addition, we would like to point out that the present method of calibration of visibility meters is mainly focused on measurement errors of the instrument's components, while the calibration in this study is focused on errors caused by the inconsistency in the measurement wavelength, which does not involve the calibration of the instrument components; and thus, suitable literature is unavailable and therefore cannot be provided.

On the point about visibility meters with multiple wavelengths, we have investigated details of commonly available commercial visibility meters and we were unable to find a multi-wavelength visibility meter. Among those commercially available visibility meters that were investigated by us,

apart from single-wavelength visibility meters, there are those that use the white-light source with a continuous emission spectrum. However, these visibility meters obtain visibility by measuring changes in light intensity and they cannot give the Ångström exponent, and therefore they are different from multi-wavelengths instruments. Some instruments use a multi-wavelength light source (e.g., sun photometers), but their output data are different from those of visibility meters. Of course, it is likely that some visibility meters do use multiple wavelengths, but we did not find any observations of the Ångström exponent given by visibility meters. In summary, we think that the influence of the Ångström exponent is not considered in current visibility measurements, and the range of Ångström exponent cannot be obtained by visibility meters in use. Figure 3 shows the possible range of the observed Ångström exponent and the range of errors caused by the uncertainty of the Ångström exponent for reference.

**3. Comment 3:** Section 3 and Figure 2. The authors fix the size distributions and refractive index for four expected aerosol types, and then use these data to determine the Angstrom exponent by varying N (although it is not a function of N). The authors should vary the size distributions and refractive indices within physically reasonable bounds to determine ranges of values for their calculated Angstrom exponents (Figure 2b) for the different aerosol types. This would be a useful result in and of itself.

**3. Response 3:** Thank you for this comment. It is true that both size distributions and refractive index influence Ångström exponent. For example, Schuster et al. (2006) have discussed the effect of bimodality of size distributions on the Ångström exponent.

However, considering the purpose of Fig. 2, no changes have been made to it. This study conducts a well-designed simulation calibration of visibility meters, which proves that it is impossible to obtain a universally applicable empirical formula for q using visibility as a variable. Figure 2 shows the probability distribution function of aerosol particles used by the four groups and the calibration results. In the simulation calibration, the number concentration of aerosol particle size was changed while the size distribution and the refractive index was unchanged. The calibration is independent of each other; therefore, the conclusion for each group is that q is not correlated with visibility. For all the four groups, the obtained value of q is different from each other due to the differences in the physical and chemical characteristics of aerosol particles; thus, the obtained empirical formula for q cannot be applied as long as the physical and chemical characteristics of aerosol particles are different from that of calibration. Therefore, we conclude that it is impossible to obtain a universally applicable empirical formula for q. We did not change Fig. 2 in order not to distract the focus of this study. As for the changes in q caused by the size distribution and refractive index of aerosol particles, Fig. 3a shows the possible range of q in 10 typical regions for reference.

**4. Comment 4:** Section 4.2. The authors note, "If it is generally believed that a definite function of q and v exists…" Both q and v depend on aerosol composition, size distributions, hygroscopicity, etc. There is no believing in this relationship – there are reasons the two are related, but simplifying assumptions attempt to relate them in a way that is easy to implement.

**4. Response 4:** We appreciate the constructive comment made by the reviewer. In previous studies on visibility measurement, q was determined by an empirical formula that uses visibility as a variable. We have revised the sentence, which now reads as follows (Line 170–172): "If it is

accepted that a definite function of q and v exists, and that the empirical formula for $q = f(v_{\lambda 1})$ is correct, then Eq. (9) should be written in advance into the programmes of visibility meters whose light source wavelength is $\lambda_1$ rather than the reference wavelength $\lambda_0$, and the output visibility is no longer $v_{\lambda 1}$ but $v'_{\lambda 0}$." In the ensuing parts of the manuscript, further explanation is given to show that such an empirical formula does not exist.

**5. Comment 5:** Section 6. In addition to the schemes explored, it would be interesting to determine q from the two wavelengths used (0.87 um and 0.44 um) for every data point, and then determine AOD at a 3rd wavelength from Aeronet (not currently included) collected at the same time, to get a better idea of uncertainty in q due solely to the Aeronet measurements.

**5. Response 5:** In fact, we considered such calculations in the data analysis, but did not perform them because if q is not correlated with measurement wavelength, such calculations can better assess the uncertainty in q due to the Aeronet measurements. However, q is correlated with the measurement wavelength and the value of q calculated at a third wavelength (e.g., 1.02 μm and 0.44 μm) is originally different from the value of q calculated at the existing two wavelengths (0.87 μm and 0.44 μm), leading to the difference in the calculated AOD at the third wavelength (in this case at 1.02 μm) originally. Therefore, for the difference in calculation for AOD at a third wavelength, it is unknown whether it is attributed to the difference in the measurement wavelength or the difference in the physical and chemical characteristics of aerosol particles, and the problem becomes more complicated.

**6. Comment6:** Line 45. A statement describing the parameters that affect the Angstrom exponent would be useful here. Line 70. You should be more specific when you state, "…errors exist in the error estimates of current visibility measurements." Research grade instruments do not suffer from these same issues.

**6. Response 6:** We appreciate the reviewer's suggestion. The following sentence in Lines 159–161 was modified as follows: "In addition to the measurement wavelength, q is determined by the physical and chemical parameters of aerosol particles in Eq. (6), i.e., the refractive index and probability distribution function for the diameter of the aerosol particles, as discussed by other researchers (Schuster et al., 2006)."

**7. Comment 7:** Line 96. Define what the hygroscopic growth factor is.

**7. Response 7:** We have added a definition of hygroscopic growth factor and the modified sentence now reads as follows (Lines 100–104): "As for hygroscopic particles on which water vapour condenses, the refractive index of mixed particles can be calculated using the weighted average of the volume ratio of each composition, and the diameter of mixed particles can be calculated using the hygroscopic growth factor (GF), which is defined as the ratio of humidified particle diameter to the diameter at dry conditions (Jacobson, 2001; Chen et al., 2012; Zhang et al., 2017)."

**8. Comment 8:** Lines 139-141. The authors note, "The common conclusion reached during calibration of the four groups is that q is a constant variable independent of visibility. This is a different conclusion from previous studies, where q is determined by an empirical formula that uses visibility as a variable." Other work which assumes a dependence of q (a proxy for size distribution)

on visibility (a proxy for aerosol loading) are based on the assumption that greater aerosol loadings typically occur following atmospheric processing (e.g. Pitchford et al., 2007 JAWMA), which results in larger aerosol sizes (and therefore a change in q). This is based on ambient measurements, not simulations where size distributions can be kept constant.

**8. Response 8:** We agree with the reviewer. As stated by you, the Ångström exponent (q) in ambient air cannot remain constant due to the constant changes in the physical and chemical parameters of aerosol particles such as size distribution and refractive index. In previous studies on visibility measurement, q is determined by an empirical formula that uses visibility as a variable; however, such an empirical formula is proven to be unreliable by the simulation calibration performed in this study.

The advantage of simulation calibration is that the conclusion can be clearly presented through an absolute control of experimental conditions. First, in the simulation calibration, the number concentration of aerosol particle size was changed while size distribution and refractive index was unchanged. Under such conditions, the measured visibility changed but the value of q remained constant for all groups; therefore, we concluded that q is not correlated with visibility and it is impossible to obtain reliable values of q unless the value of q remains constant in the visibility measurement. Then, the obtained values of q were different for each of the four groups. This shows that q is not a constant value because of the differences in the physical and chemical parameters of aerosol particles. Therefore, it is impossible to obtain a universally applicable empirical formula for q.

**9. Comment 9:** Line 167. Who is "thinking" in this sentence. Please re-phrase.

**9. Response 9:** Thank you for this comment. We have revised the sentence, which now reads as follows (Lines 176–177): "Since $q = f(v_{\lambda 1})$ is considered correct for all the four groups, the difference between $v'_{\lambda 0}$ measured by the visibility meter at $\lambda_1$ and $v_{\lambda 0}$ measured at $\lambda_0$ is wrongly attributed to the measurement error." In previous studies on visibility measurement, q is determined by an empirical formula that uses visibility as a variable ($q = f(v_{\lambda 1})$); thus, it is assumed in this study that the four groups support this approach.

**References**

Biral: RWS 30-Manual, UK: available at: https://www.biral.com/wp-content/uploads/2018/05/RWS-30-User-Manual-107384.00B.pdf, 2018. Last access: 27 Jan 2021.

Optec: Model LPV-3 & Model LPV-4-technical manual, USA: available at: https://www.optecinc.com/visibility/pdf/lpv_3&4_technical_manual_rev4.pdf, 2011. Last access: 27 Jan 2021.

Schuster, G. L., Dubovik, O., and Holben, B. N.: Angstrom exponent and bimodal aerosol size distributions, J. Geophys. Res.: Atmos., 111, doi: https://doi.org/10.1029/2005JD006328, 2006.

Vaisala: Transmissometer LT31-Datasheet, Finland: available at:

https://www.vaisala.com/sites/default/files/documents/LT31-Datasheet-B210416EN-E.pdf, last access: 2018. Last access 27 Jan 2021.

WMO: The First WMO Intercomparison of Visibility Measurements: Final Report (D.J. Griggs, D. W. Jones, M. Ouldridge and W.R. Sparks) (WMO/TD-No. 401), World Meteorological Organization, Geneva, Switzerland, 1990.

---

## Author Comment (AC2) · 17 Feb 2021

Response to reviewers for manuscript amt-2020-415: **Ångström exponent errors prevent accurate visibility measurement**

We thank the editorial team and the reviewers for your valuable inputs in enhancing the quality of our manuscript and appreciate the opportunity provided to revise it for publication in *Atmospheric Measurement Techniques*. We are happy to submit our point-by-point responses to the reviewers' comments and suggestions. The reviewers' comments/suggestions are in black. Our responses are in red. All the suggested changes have been incorporated in the manuscript and are outlined here.

**General Comments:**
The paper presents theoretical background of visibility measurements and points out systematic errors in essentially all visibility measurements. The theoretical background of the errors is pointed out on lines 34 – 64 of the discussion paper. The explanation is convincing, I really learned new things in reading it. It is obvious that there are systematic errors in visibility measurements worldwide. The topic is definitely important, not only to the scientific community but also to a wider audience: visibility measurements for instance at airports, harbours and at sea are relevant to practically everybody.
The paper is important and basically well written and I can recommend publishing it in AMT. However, before that I wish you would do some modifications.
**General response:** We thank the reviewer for the valuable review and encouraging comments.

**1. Comment:** First, I can see in your Fig. 2 that there were bimodal size distributions in your simulations already but you did not really pay any attention to it. Bimodality of size distributions have a strong effect on the Ångström exponent, see, e.g.,
Schuster, G. L., Dubovik, O., and Holben, B. N.: Angstrom Exponent and Bimodal Aerosol Size Distributions, J. Geophys. Res., 111, D07207, https://doi.org/10.1029/2005JD006328, 2006.
There are a lot of references in that paper and there are a lot of refs to it that discuss this matter. Among other things it shows that the Ångström exponent often varies with wavelength. And with the ratio of coarse and fine particles. How do these affect your results?

**1. Response 1:** Thank you for this valuable comment. We have added the reference and explanation. In this study, Eq. (6) specifies the determining variables of the Ångström exponent (q), which includes size distribution. The following sentence in Lines 159–161 to "In addition to the measurement wavelength, q is determined by the physical and chemical parameters of aerosol particles in Eq. (6), i.e., the refractive index and probability distribution function for the diameter of the aerosol particles, as discussed by other researchers (Schuster et al., 2006)."
Figure 2a shows the probability distribution function of aerosol particles used by the four groups in the simulation calibration, and Fig. 2b presents four different values of q. In this study, the design of the simulation calibration reflects the influence of size distribution on q, where the differences in size distributions of aerosol particles between groups caused the differences in the values of q given by the four groups. The aim of the simulation calibration is to reveal that it is impossible not only to obtain reliable Ångström exponents but also to determine the visibility error caused by erroneous values of q using current visibility measurement methods. We did not change Fig. 2 in order not to distract the focus of this study. As for the changes in q caused by the size distribution and the

refractive index of aerosol particles, Fig. 3a shows the possible range of q in 10 typical regions for reference.

The volume ratio of coarse and fine particles is one of the parameters that describes the size distribution of aerosol particles. It seems a bit difficult to discuss this problem using the ratio of coarse and fine particles. Additional calculations are provided below for reference. Figure 1 shows that wavelength and refractive index have a complex effect on the extinction of aerosol particles, and therefore, the calculated values of q vary over a wide range. It is clear from Fig. 1 that the extinction coefficient of fine particles is more sensitive to size distribution and refractive index than that of coarse particles. Considering the contributions of coarse and fine particles in the atmosphere to the extinction coefficient, we consider that fine particles might determine the value of q rather than the ratio of coarse and fine particles in ambient atmosphere. Of course, further work needs to be done to support such a claim.

[Figure]

**Figure 1: Relationship between the mass extinction coefficient (MEC) and the diameter of aerosol particles with different indices at a wavelength of (a) 440 nm and (b) 870 nm. The left panel shows the influence of the real part of the complex refractive index on MEC with the fixed imaginary part at 0.79. The right panel shows the influence of the imaginary part of the complex refractive index on MEC with the fixed real part at 1.95.**

**2. Comment 2:** L40 "Obviously, WMO believes ...". WMO is not a person, it cannot believe

anything. Rewrite.

**2. Response 2:** We have revised the sentence in lines 37–39, which now reads " WMO claims that "visibility and MOR should be equal" if the influence of the contrast threshold can be excluded, which implies that the choice of measurement wavelength of the light source will not affect the measurement of visibility."

**3. Comment 3:** L42 "Ångström indicated the spectral dependence of visibility as early as 1929 (Ångström, 1929) …".

I was wondering, whether Ångström really wrote about visibility and acquired a copy of the original paper. I was right. There is not even the word "visibility" in the whole paper. There is the wavelength dependency of absorption and transmittance and the derivation of the exponent that was later called the Ångström exponent. So, transmittance depends on the Ångström exponent and visibility depends on transmittance according to your Eq. (1). Your Eq. (2) follows from these but it is not given by Ångström (1929)

**3. Response 3:** Thank you for pointing out this lapse. The sentence has been modified as follows: "However, Ångström indicated the spectral dependence of the extinction coefficient as early as 1929 (Ångström, 1929). Combining the inverse relationship between visibility and extinction coefficient (Eq. (1)), we obtain:"

**4. Comment 4:** L59 "…opinion of the WMO …". again, WMO is not a person, it does not have opinions. Rewrite.

**4. Response 4:** We thank the reviewer for this useful comment. We agree that we should have phrased these points differently and not focus on WMO, but rather on specific conclusions presented in the reports by WMO. Similar to our response to Reviewer 1, in lines 55–73, we have rewritten as follows: "The Guide to Instruments and Methods of Observation (WMO, 2018) cites the intercomparison of visibility measurements (Middleton, 1952; WMO, 1990), where the difference in MOR and visibility by day is attributed to the difference in the contrast threshold. The conclusion reached in this guide states that "Visibility and MOR should be equal if the observer's contrast threshold is 0.05 (using the criterion of recognition) and the extinction coefficient is the same in the vicinity of the instrument, and between the observer and the objects." This conclusion implies that there is no need to consider a deviation from the true values of visibility measured at a wavelength of $\lambda_0$ ($v_{\lambda 0}^{\mathrm{T}}$) when converting the visibility measured at $\lambda_1$ ($v_{\lambda 1}$) to $\lambda_0$ ($v_{\lambda 0}$) using Eq. (3), i.e., the errors in the values of q are negligible. Therefore, visibility meters with different measurement wavelengths can obtain consistent visibility measurements and the following statements are true: (1) the measurement wavelength of visibility meters can be arbitrarily selected because the visibility measured at any wavelength can be mutually converted; (2) the reference visibility can be artificially defined, such as the MOR, because the reliability of visibility data will not be reduced by converting the visibility measured at various wavelengths into the reference wavelength; and (3) multiple visibility benchmarks such as MOR and meteorological visibility by day can be used simultaneously, in the same way that units such as grams and kilogrammes can be completely substituted to measure mass.

In fact, existing methods of visibility measurement using visibility meters with different wavelengths currently in use are formulated under the premise that the WMO's recommended

approach is adequate. For example, the light source of Biral RWS-30 is at 850 nm (Biral, 2018), that of Optec LPV-3 is at 550 nm (Optec, 2011), and Vaisala LT31 uses a white light source (Vaisala, 2018). However, this guide does not explain why the MOR is consistent with human observations of visibility when the contrast threshold is the same. In other words, no theoretical basis is provided to prove that reliable values of q can be obtained in the visibility measurement."

**5. Comment 5:** The Eq. (7) is strange, I have never seen it in this form. First, what do you mean with "water particles"? Does it mean humid particles or pure water droplets? Further, GF is generally defined as the ratio of humid and dry particle diameters. So, how was Eq. (7) derived? If it was used like it is written now, then you should correct it and repeat your simulations with a corrected one.

**5. Response 5:** Thank you for pointing out this error. We agree that the words "water particles" were inappropriate. We have now changed them to "pure water". We have added a definition of hygroscopic growth factor and the modified sentence now reads as follows (Lines 102–105): "As for hygroscopic particles on which water vapour condenses, the refractive index of mixed particles can be calculated using the weighted average of the volume ratio of each composition, and the diameter of mixed particles can be calculated using the hygroscopic growth factor (GF), which is defined as the ratio of humidified particle diameter to the diameter at dry conditions (Jacobson, 2001; Chen et al., 2012; Zhang et al., 2017)."

There was a missing superscript in Eq. (7), and we have now corrected the typographic error. Fortunately, the correct equation was used in the calculation. The correct form of Eq. (7) is

$$m = \frac{m_A + m_W(GF^3 - 1)}{GF^3}$$ , and the equation is derived in the following manner:

$$m = \frac{m_A V_A + m_W V_W}{V_A + V_W}$$

$$= \frac{m_A \cdot \pi / 6 \cdot D_A^3 + m_W \cdot \pi / 6 \cdot [(D_A \cdot GF)^3 - D_A^3]}{\pi / 6 \cdot (D_A \cdot GF)^3}$$

$$= \frac{m_A + m_W(GF^3 - 1)}{GF^3}$$

**References**

Biral: RWS 30-Manual, UK: available at: https://www.biral.com/wp-content/uploads/2018/05/RWS-30-User-Manual-107384.00B.pdf, 2018. Last access: 27 Jan 2021.

Optec: Model LPV-3 & Model LPV-4-technical manual, USA: available at: https://www.optecinc.com/visibility/pdf/lpv_3&4_technical_manual_rev4.pdf, 2011. Last access: 27 Jan 2021.

Schuster, G. L., Dubovik, O., and Holben, B. N.: Angstrom exponent and bimodal aerosol size distributions, J. Geophys. Res.: Atmos., 111, doi: https://doi.org/10.1029/2005JD006328, 2006.

Vaisala: Transmissometer LT31-Datasheet, Finland: available at: https://www.vaisala.com/sites/default/files/documents/LT31-Datasheet-B210416EN-E.pdf, last access: 2018. Last access 27 Jan 2021.

WMO: The First WMO Intercomparison of Visibility Measurements: Final Report (D.J. Griggs, D. W. Jones, M. Ouldridge and W.R. Sparks) (WMO/TD-No. 401), World Meteorological Organization, Geneva, Switzerland, 1990.